# Supportive care needs and challenges experienced by women diagnosed with breast cancer in Kumasi, Ghana: A qualitative exploratory study

Abigail Owusu Sekyere[1], Merri Iddrisu[2], Hadiru Iddris Mumuni[3], Kennedy Dodam Konlan [ID][2]*

1 Ghana College of Nurses and Midwives, Accra, Greater Accra Region, Ghana, 2 Department of Adult Health, School of Nursing and Midwifery, University of Ghana, Legon, Greater Accra Region, Ghana, 3 School of Nursing and Midwifery, University of Health and Allied Sciences, Ho, Volta Region, Ghana

* kdkonlan@ug.edu.gh, kennedy.konlan@gmail.com

## Abstract

### Introduction

Globally, breast cancer (BC) remains the main cause of illness and death among women. A diagnosis with BC can be emotionally devastating and draining and these could predispose women diagnosed of BC to numerous psycho-emotional challenges.

### Aim

We explored the supportive care needs and challenges experienced by women diagnosed with breast cancer in Kumasi, Ghana.

### Methods

We used an exploratory descriptive design in which qualitative data via in-depth interviews was collected from fifteen women diagnosed with BC and receiving care at the Komfo Anokye Teaching Hospital in Kumasi, Ghana. A pre-tested, semi-structured interview guide was used for the data collection. The interviews were recorded using an audiotape and the audio files were transcribed verbatim. Thematic analysis was carried out with the aid of Nvivo 10.0.

### Results

The analysis of the transcripts of interviews generated the following themes regarding supportive care needs: information needs, psychological needs and challenges experienced post BC diagnosis. On the information needs, we identified the following sub-themes; in adequate patient-specific information, challenges of disease disclosure and inadequate information on the availability of supportive care services.

**Data availability statement:** All relevant data are within the manuscript and its Supporting Information files.

**Funding:** The author(s) received no specific funding for this work.

**Competing interests:** The authors have declared that no competing interests exist.

**Abbreviations:** BC, Breast Cancer; KATH, Komfo Anokye Teaching Hospital; KBTH, Korle Bu Teaching Hospital.

Regarding the psychological needs, the following sub-themes were identified: Loss of positive attitude to life, feelings of alienation and the need for moral support. On the main theme of challenges experienced by the women post BC diagnosis, the following sub-themes were identified; financial and geographical constraint in accessing care for BC, formal referral bureaucracy hindering access to care for BC and broken extended family system in Ghana.

## Conclusion

Women diagnosed with BC in Ghana are psychologically and emotionally drained and they do not have adequate information on supportive care services available to help them cope after diagnosis. We recommend that psychological care and counselling be integrated into BC care using clinical psychologist or nursing staff with specialization in counselling and psychotherapy. We further recommend that regular information sessions are instituted at the various outpatient departments providing care for BC patients by nurse managers to provide women diagnosed with BC with information about supportive care services available.

## Introduction

Breast cancer (BC) is the most prevalent type of cancer diagnosed among women each year and the one that kills the most people [1–8]. Approximately one-fourth (25%) of all cancer cases occurring in women are attributed to BC [7–10]. In Ghana, it has been observed that out of approximately 2,900 BC cases reported annually, over 10% of these individuals die from the disease [10]. As a result, BC has emerged as the leading cause of cancer-related deaths among Ghanaian women, as noted in a study by Adam and Koranteng [11]. Recent studies [10–14] suggest a consistent rise in the incidence of BC in Ghana, particularly in the past decade. Presently, BC accounts for approximately 16% of all cancer cases in Ghana and is the most prevalent cancer among females, with an estimated rate of 76 cases per 100,000 Ghanaian women [7,12,14].

Receiving a BC diagnosis could place serious psychological and emotionally challenges on the patients particularly in culturally sensitive societies like those in Ghana [13,14]. While the burden of BC increases in Ghana and treatment for it advances, there have been over concentration of research on clinical treatment rather than the supportive care needs of patients [14]. BC is a disease most patients in Ghana, due to cultural factors [12–14], would prefer keeping it limited to only close people rather than larger social groups like the religious groups, trade groups, etc due to the associated stigma with it [1,8,13]. A BC diagnosis is an emotionally devastating and potentially life-threatening disease which could trigger suicidal ideations [12,14]. The emotional and psychological strain of BC calls for supportive care measures to improve patients' quality of life [13,14]. The literature [1,12,14] reports that the chances of survival for patients with BC depend on various factors including the quality of professional and supportive care available to the patients.

Supportive care among BC patients refers to additional care provided to BC patients which focuses on tackling the psycho-emotional concerns of the patients and putting in place measures to assists the patient cope with BC and the effects of the various treatment options [9,14]. Worldwide, several studies [2,5,8,9,14] have indicated that providing supportive care to patients with chronic conditions like BC plays a crucial role in enhancing treatment outcomes. By offering such support, the psychological and emotional stress stemming from BC diagnosis and treatment is significantly reduced, leading to improved emotional well-being for these individuals and better coping as well as improved quality of life [8,9]. It has been established that one of the surest policy and care interventions that could possibly improve the quality of care of BC patients is incorporation of supportive care [9,13,14]. Even though a recent study in Ghana [4] looked at supportive care needs in women with BC, the findings were inconclusive and focused on generating quantitative data with no opportunity to obtain in-depth data from the perspectives of the patients living with BC. It is hypothesized that supportive care greatly improves treatment outcomes for BC patients in other parts of the world [9,12] but this has not been extensively explored in most countries in sub-Saharan African, a region with under-resourced health care systems. Hence, this study explored the supportive care needs and challenges experienced by women diagnosed with BC in Kumasi, Ghana.

## Methods

### Study design

We adopted a descriptive exploratory design in which qualitative data was collected from the participants. This design was appropriate because it enabled the researchers to obtain in-depth perspectives from the participants about the phenomenon under study [15].

### Study setting

This study was conducted at the Oncology Directorate of the Komfo Anokye Teaching Hospital (KATH). KATH is the second-largest tertiary hospital in the country, chosen due to the availability of comprehensive cancer services, including a radiotherapy centre, as well as a significant number of women diagnosed with BC seeking treatment at the facility. KATH is a teaching hospital with 1200 beds, serving as a tertiary-level referral centre and affiliated with Kwame Nkrumah University of Science and Technology (KNUST). The KATH has an oncology centre and provides oncology healthcare services to patients residing in the Ashanti region. Additionally, patients from the Brong Ahafo, Bono East, Ahafo, Western, Western North, Central, Eastern, parts of the Volta and Oti Regions, as well as the five Northern regions, also seek treatment at this facility and has supportive care services available for patients.

### Study population and sample size determination

The study participants included women diagnosed of BC and receiving care at the Oncology Directorate of the KATH.

In this study we used a purposive sampling technique in which fifteen (15) women diagnosed with BC were sampled to take part in the study. The sample size was determined based on the principle of data saturation and this was achieved after the fifteenth (15th) interview. Data saturation was determined when we observed that no additional information was being elicited from the participants after the twelve interviews, so we added three more interviews and observed that no new information was being elicited.

During the data collection, the team approached about thirty-six women who met the inclusion criteria and twenty-two of them agreed to participate in the study. We then informed them about the data collection process, data redundancy and data saturation. All the twenty-two potential participants consented to take part in the study and none of them withdrew their consent. However, we observed that no additional information was being elicited from the interviews after the twelve interview hence data redundancy and we further added three additional interviews and observed data saturation at the

conclusion of the fifteenth interviews and so we informed the other seven women who had consented to be part of the study of this development and hence no need for additional data collection.

## Inclusion criteria

We included women above eighteen years of age diagnosed with BC for over one year who were receiving biomedical treatment (hospital-based treatment) at the study site. The age limit was chosen due to the legal age for consent in Ghana which is eighteen years. All the participants were individuals with advanced BC diagnosis due to the late reporting culture for BC in Ghana [12,14].

## Exclusion criteria

In this study we excluded women diagnosed with BC who were yet to commence biomedical treatment (hospital-based treatment) and women being managed for cognitive and emotional diseases post diagnosis with BC as well as women who were physically too weak to take part in the interviews.

## Selection of participants and data collection

The Participants were selected from the Oncology Directorate of the KATH in Kumasi, Ghana. We visited the unit after obtaining the necessary ethical and institutional approval and informed the nurse managers of the unit about the study on weekdays between July and August 2023 about the study. The nurse managers granted us access to the potential study participants who were waiting at the Out-Patient Department of the Unit to see their specialists. The potential participants of sound mind, above eighteen years and who had been diagnosed with BC for over a year and receiving treatment at the facility were approached by the data collection team for recruitment and data collection.

We provided adequate information to the participants on the purpose, benefits, and risk of the study among other things to each potential participant who had met the inclusion criteria and those who consented to take part in the study were then recruited for data collection.

We used a pre-tested interview guide for the data collection. This tool was developed by the researchers and pre-tested among a similar population at the Korle Bu Teaching Hospital (KBTH) in Accra, Ghana. Attached is the said interview guide as supplementary tool (S1 File). The pre-testing was done in a pilot study at a similar facility like the study site among two women diagnosed with advanced BC and receiving hospital-based treatment. The final analysis of the data did not include the interviews from the pilot study. However, the necessary modifications to the data collection tool (S1 File) were done to ensure in-depth data would be obtained related to the objective of the study. The modifications involved the introduction of probes to elicit more responses on each question in the interview guide.

We then obtained written informed voluntary consent from each participant who expressed desire to be part of the study. The data was collected from the women with the aid of a pre-tested semi-structured interview guide in English and Twi (a native Ghanaian language). During each interview session, a digital audio recorder was used to capture audio recordings. Each interview session commenced with a set of socio-demographic questions, encompassing aspects such as age, educational level, chemotherapy cycles undergone, language proficiency, and religious affiliation. Following this, participants were prompted to share their experiences and describe their psycho-emotional needs and challenges with BC diagnosis and treatment.

The interviews were conducted in one of the consulting rooms designated for the research team at the study setting. Only the research team members were present in the closed consulting room during the interviews, and we informed the participants to feel free to give us their perspective on the study area and that all information collected was solely for research purposes and had no link to the care to be rendered to them at the study site. The interviews were conducted specifically by two of the research team members (KKD and HIM) who had extensive experience in conducting qualitative

research. During the data collection, any notable incidents or challenges were recorded in the field notes, and these were incorporated into the analysis of the data.

The interviews lasted for 35–45 minutes for each participant and the data collection was done in July to August 2023. In addition, we maintained a field notebook in which we recorded important incidents during the data collection as well as key observations. The recorded audio files in Twi language were first translated into English by an expert in Twi Language. All the interviews were transcribed verbatim. Interviews in Twi were translated into English and after the transcription we did a back translation with the language expert and the participants to ensure the transcript were a true reflection of what the participants said during the interviews.

## Quality control

During the data collection, the researchers assured the participants of strict confidentiality and encouraged them to express their views confidently as the study had no impact on the care the participants were receiving from the hospital. We adopted several measures to improve the quality of the data collected [15]. The interview guide was pretested at the KBTH in Accra, Ghana among a similar population. This exercise helped to ascertain the clarity of the questions in the interview guide and determined whether the questions sought to answer the research questions. Questions that produced narrow and restrictive responses were modified to include the words "describe" or narrate to broaden the responses and to give the participants the opportunity to speak more on the topic. During the data collection, the researchers adhered strictly to methodological rigour as described below and enhanced member checking with the participants to ensure that their views had been properly represented in the transcripts from the interviews [15]. At the end of the data collection, the audio transcripts in English were confirmed as verbatim transcripts of the audio files while those conducted in Twi language were translated into English based on the understanding of the researchers and cross-checked by experts in the Twi language to ensure it reflected the true views of the participants so as to ensure the quality of the data.

## Reflexivity

The researchers who are professional nurses in academia were mindful of the power dynamics during the study and consistently explained to the participants to feel free to express their views as the outcome of the study had no impact on the care they would receive since the researchers were not staff at the study setting and were not involved in rendering care. Also, the researchers informed the participants that the outcome of the study findings could be used to advocate for the incorporation of supportive care and that views expressed would be kept confidential during and after the study in addition to adherence to other ethical conduct expected of a study of this nature. These measures helped to reduce the power dynamics during the study and helped to improve the quality of the data.

## Data management

We ensured that all the interviews were audio recorded, and the data was kept in a folder with a password and accessible only to the researchers. We carefully transcribed verbatim all the recorded audio files from the interviews into text. We then compared the transcripts with the audio files several times to verify any discrepancies and contacted the participants where there was a need for clarification. We also ensured that the participants were given unique codes based on their position among the participants (P1, P2, P3 ….P15). Participants' confidentiality was ensured by removing all identifying attributes from the data and replacing them with pseudonyms. All identifying attributes such as contact information were stored electronically in a password-protected laptop, likewise, signed informed consent forms, field notes and other important documents were kept under lock and key and the files are only accessible to the researchers. The data was also stored on an external hard drive accessible to only the researchers to prevent data loss.

## Rigour

We adhered strictly to the tenets of methodological rigour required in qualitative studies [15–17]. To ensure credibility, the interview guide was pretested among two patients who met the inclusion criteria at the KBTH's Oncology Department, Accra, Ghana. This allowed the researchers to make necessary modifications to aspects of the interview guide that seemed not to elicit responses that were relevant to the objective of the study. Additionally, a purposive sampling technique [15] was employed to ensure only women diagnosed and being managed with BC who could give a vivid account of their experiences were involved. Probing and iterative questioning was also used to elicit responses from participants and situations where there were ambiguities in the responses; clarifications were sought from the participants. To achieve transferability, the research process was described in detail so that others can evaluate the applicability of data to other contexts and settings. Records of the transcribed interviews and the analysis, as well as the results of the study, were kept for audit trail and are in custody of the 1st author. To ensure dependability in the study, a semi-structured interview guide was used for all the interviews to ensure consistency in the line of questioning among the participants. Again, a detailed description of the study design, sampling method, data collection, and analysis were also documented. To ensure confirmability, the context of data collection was documented in a field notebook during the interviews. This enhanced interpretation of the data during analysis to reflect the exact responses of the participants. The authors also bracketed their experiences and presuppositions to avoid any influence in the interpretation and analysis of the data [16,17].

## Ethics approval and consent to participate

This study was approved by the Institutional Review Board of the KATH (Protocol reference No: KATH IRB/AP/082/23). Prior to obtaining approval from the IRB of KATH, the study proposal was sent to the Oncology Directorate of KATH to assess initial ethical issues. Following this process, an initial approval to conduct the study was obtained from the Oncology Directorate of KATH. Participants who expressed willingness to participate in the research signed informed consent forms and a copy was given to them to keep. All interviews were audio recorded with participants' permission. None of the participants declined to be recorded in any of the interviews. Permission was requested from each participant to use their verbatim quotes. None of the participants objected to this request too. However, as part of this process, participants were made aware of their right to withdraw from the process at any stage without any harm to them. They were, however, assured of every effort to protect their confidentiality and anonymity. The researchers further safeguarded the emotional wellbeing of the participants by referring each participant to a clinical psychologist after the interviews to receive psychotherapy and this contributed to maintaining the participants' emotional well-being considering the sensitive nature of some aspects of the study.

## Data analysis

We performed the data analysis concurrently with the data collection with the aid of Nvivo 10.0. The researchers adopted the thematic analysis approach to data analysis as recommended in literature [17]. The interviews that were audio recorded were transcribed word for word as soon as they were concluded. The verbatim transcripts and the field notes recorded in the diary were used in the thematic analysis. The analysis followed the procedure as outlined by Braun and Clarke [17]. The 2nd and 4th authors who have advanced knowledge in qualitative research did the coding of all the transcripts after which all the authors met to reconcile any discrepancies. The researchers read the transcripts, and field notes multiple times to become comfortable with the material and pick out notable sentences and phrases. Statements and phrases that were found to be similar were categorized together and given unique codes. The codes were then put together to generate the themes and sub-themes [17]. In describing the supportive care needs and challenges of the women post BC diagnosis, the transcripts were grouped into meaningful units to generate the initial codes that led to the development of themes and sub-themes. Direct quotations were captured in the analysis to support the themes and sub-themes generated from the data.

## Results

### Socio-demographic characteristics of the participants

A total of fifteen (15) participants were interviewed for this study. All participants were females who were receiving chemotherapy for BC at the Oncology Directorate of KATH. Out of the 15 participants, 9 participants were fluent in the local language (Twi) and 6 participants spoke English language which was their most preferred language for the interviews. The average age of the participants was 51 years. Whilst a few of the participants had either Junior High level or no education, a few of the participants were also trained up to tertiary education. Most of the participants (93%) were Christian and a few (7%) were Muslims. Table 1: Provides a Summary of the Socio-Demographic Characteristics of the Study Participants.

The study participants identified three main themes: information needs, psychological needs and challenges experienced by women post BC diagnosis. On the information needs, the participants identified the following sub-themes; patient-specific information, disease disclosure and inadequate information on the availability of supportive care services. Regarding the psychological needs, the following sub-themes were identified: Loss of positive attitude to life, feelings of alienation and the need for moral support. On the main theme of challenges experienced by the women post BC diagnosis, the following sub-themes were identified; financial and geographical constraint in accessing care for BC, formal referral bureaucracy hindering access to care for BC and broken extended family system in Ghana.

Table 2 presents the synthesis of the themes and sub-themes gleaned from the data (Table 2).

### Information needs

The study participants provided insights into the specific information needs of women who have been diagnosed with BC. This need encompassed various aspects such as gaining a comprehensive understanding of BC, its causes and risk

**Table 1. Socio-demographic characteristics of participants.**

| Participant | Age | Marital status | Number of children | Average Monthly Income (Ghana Cedis (GHC)) | Years of receiving hospital-based treatment for BC | Educational level | Religion | Ethnicity | Number of Chemo-therapy cycles |
|---|---|---|---|---|---|---|---|---|---|
| P 1 | 49 years | Married | 3 | < 2000 | 2 | Tertiary | Christian | Ashanti | 3 cycles |
| P 2 | 40 years | Divorced | Nil | <2000 | 4 | Tertiary | Christian | Bono | 5 cycles |
| P 3 | 56 years | Widowed | 2 | <1000 | 2 | No education | Christian | Ashanti | 4 cycles |
| P 4 | 65 years | Married | 4 | 3000 | 3 | Junior High | Christian | Fante | 2 cycles |
| P 5 | 36 years | Single | Nil | <2000 | 3 | Senior High | Christian | Ashanti | 3 cycles |
| P 6 | 50 years | Divorced | Nil | 2000 | 3 | Junior High | Christian | Ahafo | 4 cycles |
| P 7 | 33 years | Single | 1 | 1500 | 2 | No education | Christian | Ashanti | 2 cycles |
| P 8 | 58 years | Divorced | 2 | <1000 | 2 | Junior High | Christian | Mamprusi | 3 cycles |
| P 9 | 52 years | Widowed | Nil | >2000 | 2 | Tertiary | Christian | Fante | 2 cycles |
| P 10 | 48 years | Divorced | Nil | <1000 | 2 | No education | Christian | Ashanti | 5 cycles |
| P 11 | 67 years | Divorced | 1 | <1000 | 2 | No education | Christian | Ga | 3 cycles |
| P 12 | 38 years | Single | Nil | 1800 | 3 | Tertiary | Christian | Wrawra | 2 cycles |
| P 13 | 72 years | Widowed | 6 | <1000 | 2 | No education | Muslim | Dagomba | 2 cycles |
| P 14 | 49 years | Married | Nil | 2000 | 2 | Senior High | Christian | Bono | 3 cycles |
| P 15 | 53 years | Single | Nil | 4000 | 2 | No education | Christian | Ewe | 5 cycles |

**Table 2. Themes and sub-subthemes.**

| No. | Themes | Subthemes |
|-----|--------|-----------|
| 1. | Information needs | • Patient-specific information<br>- Understanding the disease process<br>- Understanding the care process<br>- Sources of Information<br>• Challenges of disease disclosure<br>• Inadequate information on the availability of supportive care services |
| 2. | Psychological needs | • Loss of positive attitude to life<br>• Feelings of alienation<br>• Need for moral support |
| 3. | Challenges experienced by women post BC diagnosis | • Financial and geographical constraint in accessing care for BC<br>• Formal referral bureaucracy hindering access to care for BC.<br>• Broken extended family system |

factors, dispelling misconceptions associated with the disease, and taking proactive measures to seek accurate and relevant information.

## Patient-specific information

According to the participants, obtaining the appropriate information would contribute to a better comprehension of the nature of cancer itself. This would be best appropriate if conveyed to them in their own dialect and in easy-to-understand words free of medical jargon.

## Understanding the disease process

The participants advocated the need for them to be given special information in their own language which will help them understand the disease better. Some of the participants' narratives are stated below:

*"I think we need to know much about the disease from the real medical experts."* (Participant 08)

*"We need specific information on the disease particularly in our native language to help us understand the disease better."* (Participant 01)

*"I feel there is the need for better information on the condition so that we can understand what has occurred in our body, I think they should take time to explain more."* (Participant 06)

*"When I was first diagnosed, my family disowned me because they were afraid, I would infect them due to lack of information. My own Landlord also sacked me from his house because I was going to infect him and his family. It took the nurses and doctors here to call some of my family members here to explain things to them. We need information on the disease."* (Participant 10)

Another participant also expressed that:

*"Initially, my close relations thought my BC was infectious, so no one wanted to associate with me. It took the efforts of the nurses here to disabuse their minds on that before I received the needed support at home."* (Participant 03)

**Understanding the care process**

The study indicated a general lack of understanding about the care processes associated with BC among women who have been diagnosed with BC. The participants expressed a consensus regarding the importance of being provided with information about the available treatment options, including the purpose of the treatment and the prognosis or expected outcomes of their specific treatment regimen. Some of the participants' narratives are stated below:

*"We need to know what is available in terms of treatment. No matter how good or bad the disease has progressed, I believe that something can be done. So, I think that communication should be made available to us."* (Participant 15).

*"I prefer the hospital staff give us information on how to care about us in a language I speak instead of just working on us without information."* (Participant 02)

*"Some of the staff don't tell us anything at all, they don't explain what is going to be done on you. This is not helpful at all."* (Participant 11)

Specifically, participants highlighted the importance of receiving explanations and guidance related to nutrition, wound care, and exercise to help manage potential complications and promote overall well-being.

A participant shared her experience:

*"At the start of my treatment, I was given much education about the side effects of the treatment and that was very helpful. At least it was expected so it didn't take me by surprise when I started feeling that way. The nurses also gave me much information about the disease process, that is one, and how to cope with it in terms of diet, in terms of how to care for my wound, in terms of physical exercises or physical activity. And how to cope with any other things that will come or any other complications."* (Participant 01)

Additionally, the expenses related to receiving treatments were identified as additional financial considerations that women need to be considered when providing information by health professionals to patients diagnosed with BC. It is crucial to inform these women early in their illness journey about the potential costs associated with managing their condition as narrated by a participant:

*"Honestly, I was not told in detail that the treatment was this expensive. I think that information must come very early in the treatment. Even though they mentioned it, the details were not provided, which made it difficult to plan. In the middle of the treatment, I realized it was costly which sometimes even gets increased during treatment. So much money is involved in terms of investigations and treatments."* (Participant 09)

The participants also reported that one-off discussions about their condition with their treatment team were not sufficient, highlighting the need for ongoing communication and support.

A participant shared her story:

*"If there is anything I have been pleased with, it is how the health workers here talk to us. They treat us with dignity contrary to what we see in most Ghanaian hospitals. They talk to us well and are willing to answer all questions for us. What I would recommend is that the discussions should not be one-off but regular and in the local dialects. We have a lot of issues bothering us. Not all of us can come to them. Therefore, I suggest they also try to initiate the conversation. That will help."* (Participant 06)

## Sources of information

The study findings indicated that women with BC relied on various sources of information, including oral information, written materials, audio-visual resources (such as videos), the internet, healthcare providers (doctors/nurses, allied health professionals), and other BC survivors.

Some of the participants said:

*"It has to be verbal…verbal, I think sinks [in] more, especially many of us are not educated or do not understand the language of the health workers."* (Participant 08)

*"Sometimes they give us materials that we can't read, and this makes it difficult for us."* (Participant 13)

*"We share information among ourselves especially those who have experienced this condition for a long time, this way, we get real life information and experiences."* (Participant 07)

*"At times I try to read about the disease online using my phone and sometimes too, some of the health workers talk to me about the disease but this is often not regular."* (Participant 14)

## Challenges of disease disclosure

The participants stated that sharing their problems with friends, family and even their religious leaders was a good way of seeking emotional or even financial support. However, they claimed these support structures also sometimes breached the confidentiality protocols by exposing the details of their sickness to others. As a result, many participants shared that they preferred keeping their sickness and all the challenges thereof to themselves rather than sharing with family, friends and their religious groups.

A participant commented:

*"I am a believer of the adage that says, a problem shared is a problem solved but from experience this is not always the case. I attempted sharing the details of this sickness with my pastor and his wife. I expected so much support from them especially in terms of comforting me and regularly praying with me. Yes, that happened but all the details I shared with them were out in less than 2 weeks. I felt betrayed and disappointed. So, as I sit here, my issue will never be discussed with anyone except my husband and kids."* (Participant 11)

Another participant stated:

*"Sometimes you go and share your problems with your pastor hoping for support but before you realise, they are using your situation to preach." (Participant 03)*

Yet another participant stated:

*"The pastors at times share the information with the leaders of the church and they equally spread it around the church members and the community, so before you know it, everyone knows about your situation."* (Participant 07)

## Inadequate information on the availability of supportive care services

Several participants expressed a little or no awareness regarding the available services and facilities to address their psychological needs related to their condition and treatment.

Some of the participants' quotes were:

*"We don't know the available services we can use to help us manage our emotional problems with this sickness; everything is strange to me."* (Participant 10)

*"The personnel here assume we know everything and where to find help, yet I have no idea the kind of services that can help me emotionally with the worries."* (Participant 02)

*"This hospital staff are busy doing their work and appear not to have time to tell us which services are available to help us reduce our fears, they are too concerned about what they do and don't care much about us."* (Participant 13)

*"We need more information about the various supportive services. I know the nurses mentioned psychologist helping me deal with other aspects of the disease. I don't know anything about that service. There may be more. We need more information on the psychological care."* (Participant 08).

Another participant also mentioned that:

*"Also, there is limited information about where to get clinical psychologist. I was referred to one but as at now, I can't even find the person to book an appointment with him or her."* (Participant 09)

## Psychological needs

The study participants raised issues relating to psychological symptoms which affected them and the need to address these issues to ensure optimal personal control over these psychological symptoms and to maintain a positive attitude. The data under this theme was sub-categorized into the following sub-themes: loss of positive attitude to life, feelings of alienation and need for moral support.

### Loss of positive attitude to life

The loss or alteration of a breast often leads to feelings of reduced femininity, which the participants claimed contributes to the feelings of depression.

Some of the participants narrated as:

*"Psychologically, I loss hope immediately I heard about my disease. I knew I was going to die because no one survives cancer [patient said this amidst tears]. Honestly, I am unstable, I am not sound again because I am always thinking about what will happen next to me."* (Participant 06).

*"This sickness has taken life out of me, I feel I have lost everything."* (Participant 01).

*"The issue has affected me and my mind such that I can't think well about myself and my life, I have lost everything in this life."* (Participant 03).

### Feelings of alienation

The study findings revealed that women diagnosed of BC felt alienated from life and they claimed they needed support to manage their feelings which included shame. The participants claimed that the sickness made them hopeless in life, and they felt withdrawn from society. These concerns of the participants point to the psychological and emotional effects of

being diagnosed with BC and the need to put in place psycho-emotional therapies to tackle these concerns. Some participants' quotes are below

> *"This sickness brought me shame and loss of hope. The odor from the wound made me shy as I lost hope of healing. Unfortunately, everyone avoids you and no one wants to love you or get close to you."* (Participant 15)

> *"I have lost hope and the change in my breast makes me shy, there is no one to talk to and it seems no one loves me."* (Participant 11).

> *"I have become isolated in life with no one to lean unto to even cry, no one loves me. It is like I have been cursed. I look like a man without my breast"* (Participant 08)

Many participants shared their embarrassment and distress caused by the unpleasant scent emanating from their fungating tumor, leading them to devalue themselves in public settings.

> *"My lover has left the bedroom and now sleeps in the hall because of the smell; you have no idea what I go through at home."* (Participant 07)

> *"You cannot even go for social events due to the unpleasant scent, and this makes me lonely, depressed and wishing death comes quickly."* (Participant 05)

Thus, this study observed that when patients diagnosed with BC are left alone, it negatively impacts them. As emphasized by one participant, it is essential to support their quality of life by respecting and honoring their wishes and decisions, including incorporating their routines and rituals into their care. A participant shared her story:

> *"I think we need to be listened to more often and ensure our concerns are taken care of in the care process. Look, when I am alone, so many things come to my mind. Sometimes I feel the health workers are too busy to listen to us. I know the patients are many, but they should also have one-on-one meetings with us. In one of my visits where a nurse spent almost two hours talking to me and encouraging me, it helped me so much. She gave me her number to call her when anything is bothering me. Trust me, it has really calmed me down than before. We need such things routinely as part of our care."* (Participant 13)

**Need for Moral Support.** The participants stated that the emotional consequence of the disease calls for moral support. They emphasized that this type of support enables them to live their lives to the fullest. The participants claimed their sickness required persistent encouragement to face the illness and its unique treatment requirements. They claimed that they needed to be constantly assured that the sickness was manageable to reduce their fears and anxiety. This finding enjoins nurses to consistently provide reassurances to patients with BC particularly on the efficacy of the treatment options available for BC care.
A participant commented:

> *"We need reassurance. Assurance in terms of the fact that the disease can go completely and enjoy [a] normal life. We need to be encouraged to carry on with our daily activities so long as our energy can carry us. So, I think basically those are the major issues, and then, well, for our part of the world, we need reassurance that cancer is not infectious, it's not due to something we've done in the past that has resulted in the sickness."* (Participant 09)

Other participants also shared their experiences:

> *"Emotionally we are affected, and so we need someone who could or who is always there to support us emotionally, not only the healthcare givers, but [I'm] talking about the family because some of us come and go alone without support and no one to talk to at home."* (Participant 06)

*"hmmmmmm…sometimes, you just need someone to cry on or lean on when emotionally drained due to the sickness."* (Participant 04).

*"I feel overwhelmed emotionally and I need someone to comfort and provide some support in this critical time of my life."* (Participant 01)

### Challenges faced by women post diagnoses of BC

The participants identified several challenges faced by them post diagnosis of BC. These included financial and geographical access to care, formal referral bureaucracy and broken family systems which failed to provide support.

### Financial and geographical constraint in accessing care for BC

The geographical distance between the participants' residences and the place of receiving health services acted as a barrier to accessing allopathic care. In addition, the requirement of finances to seek health care was identified as a major barrier as well.
Some of the quotes from the participants in support of this fact are stated below:

*"I travelled from Bawku to come for the services here and I spent so much on transport and food."* (Participant 08)

*"Most of us are from far places and spend so much travelling time and money to come to the hospital and no one cares how you come to the hospital and where you sleep when at the hospital."* (Participant 04)

*"Sometimes I miss review days because the hospital is far, and I don't have money always to come for review."* (Participant 06)

### Formal referral bureaucracy hindering access to care for BC

The participants claimed that health Care for BC is not decentralized in Ghana. Additionally, most of the services for BC required a formal referral, and the absence of such referrals or introduction to the existing health and supportive care services prevented women diagnosed with BC from utilizing them. Some of the participants' narratives are stated below.

*"You cannot just get up and come here, unless by a referral and this makes things difficult for us because it is not easy to get a referral here."* (Participant 09).

*"I went to see several health workers in my region before I was finally referred here to be diagnosed of this disease, and it takes time before you get the referral letter to come here."* (Participant 12)

*"There is a lot of paperwork and stress before you finally get the chance to meet the specialist."* (Participant 03)

*"Even if you have to see a clinical psychologist, you need a letter or referral and getting this letter is very difficult and sometimes bureaucratic with a lot of bottlenecks."* (Participant 07)

### Broken extended family system

The participants explained that most of their supportive care needs were provided by their immediate families (including spouse and children (nuclear family) rather than extended family members. The participants claimed that unlike in time past that extended family members cared much about the welfare of others, society had changed, and most people now focus on their nuclear families to the neglect of their extended families.

A participant narrated:

*"Unlike in the past where you could easily call your brother or uncle for some help, now the situation is different. They all have their families and their own battles, so I didn't want to bother them. So, most of the help is from my husband."* (Participant 01)

Another participant also mentioned:

*"When I was told I had cancer, the first person I called was my brother. He showed sympathy for a while but immediately he saw I needed his assistance in terms of being transported to hospital regularly, paying for hospital bills and drugs among others, he started pulling out. It's unfortunate but the truth is, he also has his family now, so I don't think he was willing to add my worries to his. If not my children, things would have been tough especially my means of transport to the hospital."* (Participant 04)

## Discussion

In this study, we explored the supportive care needs and challenges experienced by women diagnosed with BC in Kumasi, Ghana. The study findings revealed that women diagnosed with BC had a significant need for information about various aspects of their condition. This included details about the diagnosis itself, treatment options, potential side effects and complications, as well as how to manage them, and the overall healthcare system processes that they needed to navigate. These informational needs as identified in this study align with those reported in previous research involving similar participants [18,19]. BC diagnosis and treatment trigger a series of health-related transitions that demand the women to negotiate several changes to their roles, coping strategies, support systems, and expectations of the sickness and these require they are well-informed [14,18,20]. Women with BC therefore need access to trustworthy, accurate, in-depth information regarding the condition, standard and developing treatment modalities, and methods of coping with symptoms and side effects/complications of treatments, not just in Ghana but globally [5,18,19]. The need for information from the participants could be a result of the low educational status of the participants as the results indicate that just a few of them attended tertiary school. There is the need for nurses to design alternative measures to give education to patients in sub-Saharan Africa. Similar informational needs have been voiced by healthcare professionals for males with prostate cancer [20]. These women must be continuously informed about the disease, as cancer patients frequently have communication breakdowns brought on by their inability to comprehend the information given to them about the disease and its treatments, their unrealistic expectations, and their psychological distress (21–24). The need for information emphasis the relevance of patient education by nurses and other care professionals. Culturally appropriate information sessions must be held with patients diagnosed with BC to give them ample scientific information related to their diagnosis and management. To ensure women diagnosed with BC are aware, it is crucial to continuously provide them with information and education on the disease and its management [18].

The women also stated that they required access to a care coordinator (a medical staff member with whom they could regularly communicate about every element of their treatment). The findings from this study call for the training of BC care navigators who will coordinate the care of patients. The results also demonstrated the necessity of providing patients with continual information regarding their condition and their treatment. This is acknowledged as a crucial step in educating BC patients about their condition and the healthcare facility they will be interacting with. Contrary to other studies [7,10,21] that stated that women preferred receiving information in writing, the results in this study suggested that health workers should deliver information orally or via the use of illustrations and diagrams. This is particularly important in sub-Saharan Africa due to the low level of literacy. Healthcare professionals must involve patients in their treatment by effectively

communicating with them to learn about their needs and preferences [22]. They have to encourage people diagnosed with BC to open up about their concerns and to work to address these concerns [23].

In this study, the women expressed concerns about the fear of cancer spreading, feelings of sadness, uncertainty about their future, worries about their loved ones, hopelessness and depression arising from changes in their body image. The study reveals that patients diagnosed with BC tend to have psychological instability requiring the incorporation of psycho-social care interventions facilitated by nurse counselors, clinical psychologist, socio-behavioral therapist, among others to promote holistic care and wellbeing. According to the study, the psychological needs of the women primarily revolved around seeking meaning in their experience with the disease and regaining a sense of control over their situation which often are unmet. If these psychosocial needs are not attended to, they can escalate into substantial issues that hinder the women's ability to cope with their diagnosis, thereby affecting their overall quality of life and predisposing such women to depression and other affective states [24].

Regarding their emotional needs, the results indicated that the women diagnosed with BC sought empathy from hospital staff, loved ones, desiring to be treated as individuals rather than just cases. The study demonstrates that health workers, particularly nurses, act as a source of emotional comfort to patients. Additionally, patients need reassurance and understanding concerning their feelings about the condition. Furthermore, having the opportunity to talk to someone who had undergone a similar experience was also deemed valuable by the participants. This study shows that the previously or earlier diagnosed patients with BC act as channels for promoting emotional stability in newly diagnosed patients as they served as "peer supports". The women in this study expressed feelings of embarrassment and neglect from their families due to the disease, which made them feel unloved and overlooked. They claimed most of them depended on the encouraging words received from other patients. It is encouraged that using peer support could produce positive outcomes among BC patients. The emotional needs expressed by women with BC in this study align with those reported by women with advanced BC in a 12-country study [25] and a similar study conducted in Ghana [26]. Numerous women conveyed feelings of being misunderstood in their experiences [27] and expressed challenges in finding support groups for BC due to the shyness and stigma associated with the disease. As a result, they lacked the crucial emotional support they could receive from fellow BC survivors.

The women participating in this study emphasized the importance of healthcare professionals validating and showing sensitivity towards their feelings, while also providing reassurance. Unfortunately, healthcare providers often prioritize cancer management over holistic patient care, leading to the neglect of patient's emotional needs [27]. Obtaining support from the medical team, family, friends and fellow BC survivors will aid patients with BC effectively cope with their emotions and thus promoting emotional stability as they navigate through their cancer journey. The different needs of women found in this study highlight how crucial it is to tailor such information and education to the unique requirements of every single woman diagnosed with BC. Additionally, many patients find it difficult to request treatments or seek out health information about their illness, with many Ghanaian women with BC relying on their healthcare practitioners for information about BC [11,28]. However, there is a power imbalance between patients and medical staff in Ghanaian society, which can make it difficult for women to ask health workers for information about their disease and available therapies. In Ghana, patients and healthcare practitioners rarely consult one another while making treatment decisions because most health worker-patient contact is health worker-centered [18,28]. The study's results indicate the need for a shared decision-making approach in BC care. Actively listening to patients to comprehend their needs and preferences, and communicating accordingly, is considered a crucial component of patient-centered care [11,29,30]. Considering the diverse array of needs among women with BC in Ghana, it is essential for healthcare providers to encourage women to actively participate in seeking the information they require and engaging in decision-making about their condition and management. To accommodate various preferences, appropriate information and education materials, including face-to-face oral presentations, should be developed and provided to these women [18].

The participants also raised concerns with people keeping the details of their medical information secret. For instance, it was noted that family members and religious leaders are good sources of providing support to the women. However, this study revealed these sources of support have the potential of leaking confidential information about the women as against their wishes. As noted by other authors, this has the potential to worsen their emotions and negatively affect outcomes of their treatment [5,24,27-28–].

Aligned with the literature [10,11,28,29], the women with BC in Ghana from this study shared various barriers and challenges they encountered when attempting to use health and supportive care services. Financial constraint was a common issue, impacting the women diagnosed with BC's ability to travel to healthcare facilities and to utilize services that required payment. Additionally, many of the women lacked awareness of the support services available to them. Furthermore, accessing most of the available services required a formal introduction or referral, making it impossible for the women to access these services without such a referral.

The unavailability or limited availability of health and support services, particularly the services of clinical psychologists were viewed as a major hindrance in the management of women with BC in Ghana. Most of the participants at some point needed the services of a clinical psychologist to build them up mentally but these were mostly unavailable due to lack of information about the existence of clinical psychologists. In accessing health and support services, some of the barriers mentioned by the women in the study were the distance to service locations and the lack of assistance to reach them. Other participants did not provide a specific reason for not accessing available services. Like the findings in this study, existing literature documents various barriers and challenges faced by cancer patients and survivors in accessing supportive care services [13,14,24,30]. These include feeling too unwell, being unaware of the support services available in their local area, not receiving any introduction or referral from their oncologists, being too occupied with other responsibilities, limited access to services, lack of transportation, and unfamiliarity with the nature of support services [9,13,14].

This study found that the changing social systems in Ghana particularly the family system in Ghana which is now more concentrated on the nuclear family rather than the extended as seen in years past as an emerging barrier to care for women with BC as reported by other studies [28–30]. The extended family, which is expected to play a key role in supporting members, especially in times of chronic sickness, tends to neglect such members nowadays. As noted in this study, many of the participants did not feel encouraged to share their problems with the extended family due to their expectations or experience of low support received from them. Yet most of the participants did not have the benefit of nuclear family members as most of them mentioned they were unmarried and did not enjoy any spousal support. The changes in the family system reported in this study could be largely due to the changes in the economic dynamics in the Ghanaian society where resources are getting more limited and this makes people to focus more on their personal issues/problems and those of their nuclear families to the neglect of the issues/problems of members of the extended families [28,30]. Furthermore, urbanization and formal education are re-organizing and reshaping the family structures in most countries in sub-Saharan Africa leading to the breakdown of the extended family system concept and thus denying most people the safety nets and support provided by extended family members and yet the results revealed most of the participants were unmarried and did not enjoy spousal support. This finding corroborates the results of other past studies conducted in Ghana [5,6,28,30] suggesting that family members served as integral support systems for women diagnosed with BC in their bid to cope with the diagnosis and management.

## Limitations of the study

The study sample only includes women who are actively seeking treatment for BC, and results do not include perspectives of women who are not seeking hospital treatment for their cancer. Future studies could look at comparing the perspectives from these different populations to identify the psycho-social concerns of these varied populations

This study was conducted in a limited population of women diagnosed with BC at the KATH due to the design adopted for the study. Even though the findings give in-depth information on the subject matter, future studies should focus on using a quantitative design and a larger population to be able to generalize the findings.

Another key limitation of the study was that the participants were not limited by age as the ages of the participants ranged between 33–75 years. The differences in age could have significant emotional implications and this could have affected the findings of the study more especially the fact that the participants were females and could have emotional challenges with menopause and its hormonal imbalances. Future studies could focus on the different age brackets to ensure more robust findings.

Also, the sample size and single setting that was used for the study might limit the generalizability of findings to other regions in Ghana. However, the findings provide a basis for other future country-wide studies to improve the care of women diagnosed with BC.

Furthermore, there was a potential of power dynamics between the researchers and the participants as the researchers were trained nurses involved in the training of nurses. This could have created a situation compelling the participants to take part in the study. However, the authors tried their best to mitigate this dynamic during recruitment and data collection by emphasizing the need for the participants to voluntarily and freely decided to take part in the study and the need to maintain privacy in the reporting of the study findings. The authors further bracketed their personal experiences from the data and adhered strictly to quality control processes throughout the study.

## Conclusion

Women diagnosed of BC in Ghana are psychologically and emotionally drained and they do not have adequate information on supportive care services available to help them cope well after diagnosis.

We recommend that psychological care and counselling be integrated into BC care using clinical psychologist or nursing staff with specialization in counselling and psychotherapy. We further recommend that regular information sessions are instituted at the various outpatient departments providing care for BC patients to provide women diagnosed with BC information about supportive care services available.

## Supporting information

**S1 File. Interview guide.**
(DOCX)

## Author contributions

**Conceptualization:** Kennedy Dodam Konlan, Merri Iddrisu.

**Data curation:** Abigail Owusu Sekyere.

**Formal analysis:** Abigail Owusu Sekyere.

**Investigation:** Kennedy Dodam Konlan, Abigail Owusu Sekyere, Merri Iddrisu, Hadiru Iddris Mumuni.

**Methodology:** Merri Iddrisu.

**Project administration:** Hadiru Iddris Mumuni.

**Resources:** Hadiru Iddris Mumuni.

**Writing – original draft:** Kennedy Dodam Konlan.

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
