## [Decision Letter · Decision Letter 0]

11 Aug 2025

Dear Dr. Konlan,

Thank you for submitting your manuscript to PLOS ONE. After careful consideration, we feel that it has merit but does not fully meet PLOS ONE’s publication criteria as it currently stands. Therefore, we invite you to submit a revised version of the manuscript that addresses the points raised during the review process.

**ACADEMIC EDITOR:**
** **

We look forward to receiving your revised manuscript.

Kind regards,

Frank Kyei-Arthur, Ph.D.

Academic Editor

PLOS ONE

Journal Requirements:

3. Please include captions for your Supporting Information files at the end of your manuscript, and update any in-text citations to match accordingly. Please see our Supporting Information guidelines for more information: http://journals.plos.org/plosone/s/supporting-information .

Reviewers' comments:

Reviewer's Responses to Questions

**Comments to the Author**

1. Is the manuscript technically sound, and do the data support the conclusions?

Reviewer #1: Yes

Reviewer #2: Yes

2. Has the statistical analysis been performed appropriately and rigorously?

Reviewer #1: N/A

Reviewer #2: Yes

3. Have the authors made all data underlying the findings in their manuscript fully available?

Reviewer #1: Yes

Reviewer #2: Yes

4. Is the manuscript presented in an intelligible fashion and written in standard English?

Reviewer #1: Yes

Reviewer #2: Yes

Reviewer #1: Dear authors

Thank you for your efforts to do this research. Below are some comments and suggestions to help improve the clarity and impact of your study.

Abstract:

The abbreviation for breast cancer, commonly used as "BC" instead of "bCa" should be consistently applied throughout the manuscript.

Please add the study duration, sampling method to the Methods section.

Consider specifying the research design (qualitative exploratory) in the title to enhance clarity for the reader.

Please write the number "fifteen" in one format, either as "15" or "fifteen."

Results section: The authors have stated, "Results: The study participants identified the following themes ........" This paragraph requires revision. The themes were identified based on the participants' narratives, as it is the authors who conducted the analysis, not the participants.

Introduction section:

The introduction could be enriched by briefly discussing the cultural implications of breast cancer diagnosis in Ghana, particularly the stigma associated with it.

Please use the abbreviation "BC" for breast cancer.

Remove the heading "Aim" and integrate the study's aim into the final sentence of the Introduction.

Methods:

Expand on why Nvivo 10.0 was selected for thematic analysis, considering that newer versions exist.

Provide additional details about how translation accuracy was ensured during transcription.

Consider elaborating on how potential biases from the researchers being trained nurses were mitigated during the study.

Results:

Include a visual representation (e.g., thematic map) of the identified themes and subthemes to aid comprehension.

In the socio-demographic results, consider providing percentages for clarity, particularly concerning education and language distribution.

Some of the quotes are excessively long. I suggest summarizing them.

Discussion Section:

I suggest enriching the discussion by addressing related studies. For example, I recommend the following articles:

https://link.springer.com/article/10.1186/s12905-021-01212-9

https://link.springer.com/article/10.1186/s12905-022-01906-8

Highlight how findings from this study are novel or expand upon existing knowledge.

Address the potential influence of socioeconomic status on the participants' experiences and needs.

Discuss the implications of the broken extended family system on long-term patient support more explicitly.

Limitations:

The study acknowledges its limitations, including the narrow demographic and the absence of non-treatment-seeking participants. Please discuss how the small sample size and single setting might limit the generalizability of findings to other regions in Ghana.

References:

Ensure consistent formatting of references and check for any broken or outdated links (e.g., DOI links).

Reviewer #2: Thank you for the opportunity to consider me for this review. I have provided detailed comments for the authors’ consideration in the attached document. I recommend the manuscript for acceptance once the issues raised have been thoroughly addressed.

**Do you want your identity to be public for this peer review?** For information about this choice, including consent withdrawal, please see our Privacy Policy

Reviewer #1: No

Reviewer #2: **Yes: ** Evans Osei

---

## [Author Response · Author response to Decision Letter 1]

9 Sep 2025

University of Ghana

College of Health Sciences

School of Nursing and Midwifery

Department of Adult Health

6th SEPTEMBER, 2025

The Editor

PLOS ONE

Dear Sir/Madam,

Response to review

General comments of authors

We have addressed all the comments of the reviewers as suggested and we hope the revised manuscript meets the standards for publications.

Comments of the Reviewers

Reviewer #1: Dear authors

Thank you for your efforts to do this research. Below are some comments and suggestions to help improve the clarity and impact of your study.

Abstract:

The abbreviation for breast cancer, commonly used as "BC" instead of "bCa" should be consistently applied throughout the manuscript.

Please add the study duration, sampling method to the Methods section.

Consider specifying the research design (qualitative exploratory) in the title to enhance clarity for the reader.

Please write the number "fifteen" in one format, either as "15" or "fifteen."

Results section: The authors have stated, "Results: The study participants identified the following themes ........" This paragraph requires revision. The themes were identified based on the participants' narratives, as it is the authors who conducted the analysis, not the participants.

Introduction section:

The introduction could be enriched by briefly discussing the cultural implications of breast cancer diagnosis in Ghana, particularly the stigma associated with it.

Please use the abbreviation "BC" for breast cancer.

Remove the heading "Aim" and integrate the study's aim into the final sentence of the Introduction.

Methods:

Expand on why Nvivo 10.0 was selected for thematic analysis, considering that newer versions exist.

Provide additional details about how translation accuracy was ensured during transcription.

Consider elaborating on how potential biases from the researchers being trained nurses were mitigated during the study.

Results:

Include a visual representation (e.g., thematic map) of the identified themes and subthemes to aid comprehension.

In the socio-demographic results, consider providing percentages for clarity, particularly concerning education and language distribution.

Some of the quotes are excessively long. I suggest summarizing them.

Discussion Section:

I suggest enriching the discussion by addressing related studies. For example, I recommend the following articles:

https://link.springer.com/article/10.1186/s12905-021-01212-9

https://link.springer.com/article/10.1186/s12905-022-01906-8

Highlight how findings from this study are novel or expand upon existing knowledge.

Address the potential influence of socioeconomic status on the participants' experiences and needs.

Discuss the implications of the broken extended family system on long-term patient support more explicitly.

Limitations:

The study acknowledges its limitations, including the narrow demographic and the absence of non-treatment-seeking participants. Please discuss how the small sample size and single setting might limit the generalizability of findings to other regions in Ghana.

References:

Ensure consistent formatting of references and check for any broken or outdated links (e.g., DOI links).

Authors’ Response to Reviewer 1

We are grateful for the comments and suggestions of the reviewer and have addressed them appropriately in the revised manuscript to make same better.

Abstract:

We have re-written the abbreviation for breast cancer from “bCa” to “BC” in the revised manuscript as suggested by the reviewer.

We have added the study duration and sampling methods in the method section. This is found on page 5 of the revised manuscript under data collection.

We have stated the research design in the title of the revised manuscript as suggested by the reviewer. This can be found on the cover page of the revised manuscript.

We have re-written the number fifteen in the abstract of the revised manuscript on page 2.

Results section: We have re-written the results in the abstract of the revised manuscript on page 2 of the revised manuscript.

Introduction section:

We have stated the cultural implications of breast cancer diagnosis in Ghana , particularly the stigma as suggested by the reviewer. This is found on page 3 of the revised manuscript.

We have consistently used the abbreviation "BC" for breast cancer as suggested by the reviewer.

We have removed the "Aim" and integrated the study's aim into the final sentence of the Introduction on page 4 of the revised manuscript.

Methods:

We used Nvivo 10.0 for the data analysis as that was the available tool for the researchers as there was no funding to enable us purchase the newer versions of Nvivo.

We have provided additional details about how translation accuracy was ensured during transcription and this is found on page 7 of the revised manuscript.

We have elaborated on how potential biases from the researchers being trained nurses were mitigated during the study and this is found on page 7 of the revised manuscript on reflexivity.

Results:

We have provided further details on the socio-demographic characteristics and the table of themes and sub-themes regarding the data that was generated. We could not provide percentages due to the design of the study. The table of socio-demographics and themes as well as sub-themes can be found on pages 10-12 of the revised manuscript.

We have reduced the length of some of the quotes that were excessively long as suggested by the reviewer.

Discussion Section:

We have enriched the discussion and incorporated the suggested studies into it as recommended by the reviewer. This is found on pages 20-24 of the revised manuscript.

We have stated in the discussion section on pages 20-24 about the uniqueness of this study and further justified the reason why this study should be published to provide in-depth data (a qualitative) perspective on the subject matter

We have addressed the implications of the broken family system and the income status of the participants in the discussion section of the revised manuscript on pages 20 to 24.

Limitations:

We have discussed how the small sample size and single setting might limit the generalizability of findings to other regions in Ghana as recommended by the reviewer. This is found on pages 25 of the revised manuscript.

References:

We have ensured completeness of the reference list as suggested by the reviewer. This is found on pages 27 to 30 of the revised manuscript.

We are grateful for the comments of the reviewer.

Reviewer 2 Report

Thank you for the opportunity to review this well-written and timely manuscript, which aims to explore the lived experiences and supportive care needs of women with breast cancer in Ghana. The topic is highly relevant, with important implications for patient-centered care in low- and middle-income countries. The paper is generally well-structured and supported by relevant literature. However, I have several suggestions to improve clarity, methodological transparency, and overall rigor.

Abstract

1. Please revise the sentence “The study participants identified the following themes”. Themes are generated by the research team based on participants’ responses; they are not directly “identified” by participants. Consider rephrasing to indicate that the themes emerged from analysis of participants’ accounts.

2. Briefly state the number of themes and subthemes generated in the study.

3. Replace “diagnosed of bCa” with “diagnosed with breast cancer” for grammatical accuracy.

4. In the keywords section, only the term “Keywords” should be in bold. The list of keywords should not be bolded.

Introduction

1. The introduction is well-supported by up-to-date literature.

2. The sentence “Recent studies [10-14] suggests of a consistent rise in the incidence of breast cancer in Ghana, particularly in the past decade” contains grammatical errors:

o “Studies” is plural; use “suggest” instead of “suggests”.

o Remove “of” in “suggests of.”

o Suggested revision: “Recent studies [10–14] suggest a consistent rise in the incidence of breast cancer in Ghana, particularly in the past decade.”

3. Clearly articulate the research gap related to supportive care needs of women with breast cancer in Ghana. Some studies have already addressed this so Please emphasize what this study adds to fill the gap and how it advances the understanding of supportive care needs in this context. This and other few studies have been done Akuoko, C. P., Chambers, S., & Yates, P. (2022). Supportive care needs of women with advanced breast cancer in Ghana. European Journal of Oncology Nursing, 58, 102142. https://doi.org/10.1016/j.ejon.2022.102142

Methods

1. Provide the full list of inclusion criteria, including age range, diagnostic confirmation, and treatment stage. If certain criteria (e.g., age limits) were not applied, explain why.

2. In the exclusion criteria, clarify what is meant by “biomedical treatment,” as some readers may not be healthcare professionals.

3. Describe in detail how participants were purposively selected, including the recruitment process.

4. Expand on the explanation of data saturation:

o Describe the criteria used to determine that saturation was reached after the fifteenth interview.

o Indicate how many women were initially contacted, how many agreed to participate, and how many withdrew after consenting.

5. For the “pre-tested interview guide,” state how many participants it was tested on and whether those pilot participants were included in the final analysis.

6. Specify the location(s) of the interviews, who was present during data collection, and whether any notable incidents or challenges were recorded in the field notes.

7. Clarify whether all authors conducted interviews or whether only certain authors were involved in data collection.

8. Consider formatting the Methods section according to the COREQ (Consolidated Criteria for Reporting Qualitative Research) checklist to ensure completeness and transparency.

Results

The results are clearly presented; however, the following additions would enhance the depth of the findings:

1. If available, provide participants’ year of diagnosis, details of their support system (formal or informal), which family members were involved, and the duration of their support.

2. Include marital status and whether participants have children, as these factors may influence supportive care needs.

3. Since financial constraints were a key finding, consider reporting participants’ approximate monthly or annual income ranges for context.

4. The themes “Psychological needs” and “Challenges experienced by women post bCa diagnosis” appear conceptually linked. Consider revising to ensure clear distinction or integration to avoid redundancy.

Minor Revisions

• In the Limitations section, correct “cancerdiagnosis” to “cancer diagnosis”.

• Correct “KATH- Komfo Anokye Teachigng Hospital” to “KATH – Komfo Anokye Teaching Hospital”.

• Spell out all acronyms at first mention in both the abstract and the main text.

Overall Assessment:

The manuscript addresses an important gap in the literature and has potential for high impact. Strengthening the articulation of the research gap, providing more methodological detail, and clarifying certain grammatical points will enhance its rigor and clarity.

Authors’ Response to Reviewer 2.

We are grateful for the comments of the reviewer on our manuscript and we have addressed all the concerns of the review.

Abstract

1. We have re-written the abstract of the study as recommended by the reviewer. We rephrase the sentences in the revised manuscript.

2. We have stated the number of themes in the revised abstract on page 2 of the revised manuscript.

3. We have replaced the suggested statement “diagnosed of bCa” with “diagnosed with breast cancer” for grammatical accuracy as suggested by the reviewer. This can be found on page 2 of the revised manuscript.

4. We have complied with the instructions of the reviewer in the keywords section by removing the parts that do not require to be bold. This is found on page 2 of the revised manuscript.

Introduction

1. We have removed the grammatical errors in the introduction of the revised manuscript as suggested by the reviewer. This is found on pages 3 and 4 of the revised manuscript.

2. We have acknowledged previous studies on the subject matter and further stated the gap in the literature warranting our study as suggested by the reviewer. This is found on pages 3 and 4 of the revised manuscript.

Methods

1. We have provided the full list of inclusion criteria in the methods section of the revised manuscript as suggested by the reviewer. This is found on page 5 of the revised manuscript.

2. We have explained the meaning of the term “biomedical treatment” used in the inclusion an exclusion criterion to provide further clarity as suggested by the reviewer. This is found on page 5 of the revised manuscript.

3. We have described in detail how participants were purposively selected, including the recruitment process as suggested by the reviewer. This is found on pages 5 and 6 of the revised manuscript.

9. We have provided information on how many women were contacted, how many agreed to take part in the study, when data redundancy started occurring and when data saturation was achieved. This is found on pages 4 and 5 of the revised manuscript under study population and sample size determination.

10. We have provided information on the pre-testing of the tool. This is found on page 6 of the revised manuscript.

11. We have stated the specific location where the interviews for the study were conducted, and this is found on page 6 of the revised manuscript. We have further provided information on page 6 as to the specific persons among the authors who conducted the interviews.

Results

The results are clearly presented; however, the following additions would enhance the depth of the findings:

1. We have provided the participants’ year of diagnosis, marital status, average monthly income as reported by the participants and whether they had children in the socio-demographic table on page 10 of the revised manuscript.

4. We have re-done the table of themes to capture the detailed information from the participants and to make the paper robust as suggested by the reviewer.

Minor Revisions

• We have done the suggested corrections in the limitations of the study as suggested by the reviewer. This is found on pages 24 and 25 of the revised manuscript.

• We have corrected the spelling of “KATH- Komfo Anokye Teachigng Hospital” to “KATH – Komfo Anokye Teaching Hospital” in the abbreviations section of the revised manuscript as suggested by the reviewer. This is found on page 25 of the revised manuscript.

We hope the revised manuscript will meet the standards for publication.

Thank you

Yours sincerely,

Dr. Kennedy Dodam Konlan

(Corresponding author)

---

## [Decision Letter · Decision Letter 1]

28 Oct 2025

Dear Dr. Konlan,

Thank you for submitting your manuscript to PLOS ONE. After careful consideration, we feel that it has merit but does not fully meet PLOS ONE’s publication criteria as it currently stands. Therefore, we invite you to submit a revised version of the manuscript that addresses the points raised during the review process.

**ACADEMIC EDITOR:**
**Abstract**

The phrase “Conclusion: Women diagnosed with breast cancer” should be made “Conclusion: Women diagnosed with BC”.The phrase “integrated into breast cancer care” should be made “integrated into BC care”

**Methods**

Under study design, the authors should kindly indicate why a descriptive exploratory design is appropriate for their study.“Setting” should be revised to “Study setting”Under Study setting, after the abbreviation “KATH” is used, all Komfo Anokye Teaching Hospital should be replaced with the abbreviation.The phrase “at the study site” should be made “study setting” for consistency.Replace the phrase “study site” with “study setting” throughout the manuscript.Under “Selection of participants and data collection”, Komfo Anokye TeachingHospital should be replaced with the phrase “KATH”.The phrase “not staff at the study settings” should be revised to “not staff at the study setting”The phrase “1st author” should be written as “1^st^ author”The phrase “post bCa diagnosis” should be replaced with “post BC diagnosis”

**Results**

The first three sentences under the result section and the socio-demographic characteristics of the participants sub-section are a repetition. The authors should kindly address it.I suggest the authors delete the statement “The face-to-face semi-structured interviews took placebetween July, 2023 to August, 2023 and was personally conducted by the researchers.” from the socio-demographic characteristics of the participants sub-section.I suggest the authors merge the information under the sub-section “results from the in-depth interviews” into the sub-themes of “information needs”, “psychological needs” and “challenges experienced by women post BC diagnosis”. This approach will enhance the flow of the results write-up.The labelling for quote “*“I feel there is the need for better information on the condition so that we can understand what has occurred in our body, I think they should take time to explain more”* is missing.Some of the quotes have full stop at the end, while others do not. I suggest the authors fill all those grammatical errors.In the abstract section, the authors used the abbreviation “BC” for breast cancer. However, in the introduction and discussion sections, the authors used the full word “breast cancer” instead of the abbreviation. Furthermore, in the results section, the authors are using the abbreviation “BC”. I suggest the authors decide on one style and use it throughout the manuscript.The authors used both British and American English. E.g., sub-categorized, decentralized and odour. I suggest the authors decide on one and use it throughout the manuscript.

**Discussion**

I suggest the authors provide a brief statement(s) about the aim of the study before discussing the results of the study.I noticed that the authors had a sub-section “Challenges faced by women diagnosed with breast cancer” under the discussion section. If the authors want to discuss the results in sub-sections, then they should create sub-sections for the two remaining objectives of the study to ensure consistency. Otherwise, they can remove the sub-section for “Challenges faced by women diagnosed with breast cancer”.Under the limitations of the study, the abbreviation for Komfo Anokye Teaching Hospital should be used.The authors have the abbreviation “KBTH”. But the abbreviation was not used in the manuscript.I suggest the authors summarise the write-up under the “ethics approval and consent to participate” sub-section. 

We look forward to receiving your revised manuscript.

Kind regards,

Frank Kyei-Arthur, Ph.D.

Academic Editor

PLOS ONE

Journal Requirements:

Reviewers' comments:

Reviewer's Responses to Questions

**Comments to the Author**

Reviewer #1: (No Response)

Reviewer #3: All comments have been addressed

2. Is the manuscript technically sound, and do the data support the conclusions?

Reviewer #1: Yes

Reviewer #3: Yes

3. Has the statistical analysis been performed appropriately and rigorously?

Reviewer #1: N/A

Reviewer #3: Yes

4. Have the authors made all data underlying the findings in their manuscript fully available?

Reviewer #1: Yes

Reviewer #3: Yes

5. Is the manuscript presented in an intelligible fashion and written in standard English?

Reviewer #1: Yes

Reviewer #3: Yes

Reviewer #1: Dear authors

Thank you for addressing my comments. I think there is a good improvement in the manuscript. I have no additional comments.

Best regard

Reviewer #3: (No Response)

**Do you want your identity to be public for this peer review?** For information about this choice, including consent withdrawal, please see our Privacy Policy

Reviewer #1: No

Reviewer #3: **Yes: ** Kofi Boamah Mensah

---

## [Author Response · Author response to Decision Letter 2]

30 Oct 2025

University of Ghana

College of Health Sciences

School of Nursing and Midwifery

Department of Adult Health

29th OCTOBER, 2025

The Editor

PLOS ONE

Dear Sir/Madam,

Response to review

General comments of authors

We have addressed all the comments of the reviewers as suggested and we hope the revised manuscript meets the standards for publications.

Comments of the Reviewers

PONE-D-25-23079R1

Supportive care needs and challenges experienced by women diagnosed with breast cancer in Kumasi, Ghana: A qualitative exploratory study

PLOS ONE

Dear Dr. Konlan,

Thank you for submitting your manuscript to PLOS ONE. After careful consideration, we feel that it has merit but does not fully meet PLOS ONE’s publication criteria as it currently stands. Therefore, we invite you to submit a revised version of the manuscript that addresses the points raised during the review process.

ACADEMIC EDITOR:

The authors should address the comments below to further strengthen their manuscript.

Abstract

1. The phrase “Conclusion: Women diagnosed with breast cancer” should be made “Conclusion: Women diagnosed with BC”.

2. The phrase “integrated into breast cancer care” should be made “integrated into BC care”

Methods

1. Under study design, the authors should kindly indicate why a descriptive exploratory design is appropriate for their study.

2. “Setting” should be revised to “Study setting”

3. Under Study setting, after the abbreviation “KATH” is used, all Komfo Anokye Teaching Hospital should be replaced with the abbreviation.

4. The phrase “at the study site” should be made “study setting” for consistency.

5. Replace the phrase “study site” with “study setting” throughout the manuscript.

6. Under “Selection of participants and data collection”, Komfo Anokye Teaching

7. Hospital should be replaced with the phrase “KATH”.

8. The phrase “not staff at the study settings” should be revised to “not staff at the study setting”

9. The phrase “1st author” should be written as “1st author”

10. The phrase “post bCa diagnosis” should be replaced with “post BC diagnosis”

Results

1. The first three sentences under the result section and the socio-demographic characteristics of the participants sub-section are a repetition. The authors should kindly address it.

2. I suggest the authors delete the statement “The face-to-face semi-structured interviews took place

3. between July, 2023 to August, 2023 and was personally conducted by the researchers.” from the socio-demographic characteristics of the participants sub-section.

4. I suggest the authors merge the information under the sub-section “results from the in-depth interviews” into the sub-themes of “information needs”, “psychological needs” and “challenges experienced by women post BC diagnosis”. This approach will enhance the flow of the results write-up.

5. The labelling for quote ““I feel there is the need for better information on the condition so that we can understand what has occurred in our body, I think they should take time to explain more” is missing.

6. Some of the quotes have full stop at the end, while others do not. I suggest the authors fill all those grammatical errors.

7. In the abstract section, the authors used the abbreviation “BC” for breast cancer. However, in the introduction and discussion sections, the authors used the full word “breast cancer” instead of the abbreviation. Furthermore, in the results section, the authors are using the abbreviation “BC”. I suggest the authors decide on one style and use it throughout the manuscript.

8. The authors used both British and American English. E.g., sub-categorized, decentralized and odour. I suggest the authors decide on one and use it throughout the manuscript.

Discussion

1. I suggest the authors provide a brief statement(s) about the aim of the study before discussing the results of the study.

2. I noticed that the authors had a sub-section “Challenges faced by women diagnosed with breast cancer” under the discussion section. If the authors want to discuss the results in sub-sections, then they should create sub-sections for the two remaining objectives of the study to ensure consistency. Otherwise, they can remove the sub-section for “Challenges faced by women diagnosed with breast cancer”.

3. Under the limitations of the study, the abbreviation for Komfo Anokye Teaching Hospital should be used.

4. The authors have the abbreviation “KBTH”. But the abbreviation was not used in the manuscript.

5. I suggest the authors summarise the write-up under the “ethics approval and consent to participate” sub-section.

:

Authors’ Response to Review

Abstract

1. We have done the rephrasing of the conclusion from “Women diagnosed with breast cancer” to : Women diagnosed with BC” in the abstract as suggested by the reviewer. This is found on page 2 of the revised manuscript.

2. We have changed the phrase “integrated into breast cancer care” should be made “integrated into BC care” in the abstract as suggested by the reviewer. This is found on page 2 of the revised manuscript.

Methods

1. We have indicated in the methods section why a descriptive exploratory design is appropriate for their study. This is found on page 4 of the revised manuscript.

2. We have revised the word “Setting” to “Study setting” as suggested by the review. This is on page 4 of the revised manuscript.

3. We have replaced the Komfo Anokye Teaching Hospital with the abbreviation “KATH” after its first usage throughout the revised manuscript.

4. We have replaced the phrase “at the study site” with “study setting” for consistency. This is found on page 6 of the revised manuscript.

5. We have ensured that under “Selection of participants and data collection”, Komfo Anokye Teaching Hospital has been replaced with the phrase “KATH”. This is found on page 5 of the revised manuscript.

6. We have replaced the phrase “not staff at the study settings” to “not staff at the study setting” as suggested by the reviewer. This is found on page 6 of the revised manuscript.

7. We have re-written the phrase “1st author” to “1st author” in the revised manuscript as suggested by the reviewer.

8. We have re-written the phrase “post bCa diagnosis” to “post BC diagnosis” as suggested by the reviewer.

Results

1. We have removed the repetitions in the results section. This is found on page 9 of the revised manuscript.

2. We have deleted the statement “The face-to-face semi-structured interviews took place between July 2023 to August, 2023 and was personally conducted by the researchers.” from the socio-demographic characteristics of the participants sub-section in the results section as suggested by the reviewer.

3. We have merged the information under the sub-section “results from the in-depth interviews” into the sub-themes of “information needs”, “psychological needs” and “challenges experienced by women post BC diagnosis” to help with information flow. This is found on page 11 of the revised manuscript.

4. We have rectified the labelling for quote ““I feel there is the need for better information on the condition so that we can understand what has occurred in our body, I think they should take time to explain more” which was missing. This is found on page 13 of the revised manuscript.

5. We have rectified the grammatical issues with the quotes as suggested by the reviewer.

6. We have ensured that the abbreviation “BC” for breast cancer used in the abstract has been used throughout the entire work for consistency as suggested by the reviewer.

7. We have corrected the grammatical issues in the manuscript and stuck to British English as suggested by the reviewer.

Discussion

1. We have provided a brief statement about the aim of the study before discussing the results of the study as suggested by the reviewer. This is found on page 20 of the revised manuscript.

2. We have removed the sub-section section “Challenges faced by women diagnosed with breast cancer” in the discussion to ensure the discussion flows consistently as suggested by the reviewer.

3. We have ensured that the abbreviation for Komfo Anokye Teaching Hospital is used in the limitation section on page 24 of the revised manuscript.

4. We have provided where the abbreviation “KBTH” was used in the manuscript. This is found on page 8 of the revised manuscript.

5. We have summarized write-up under the “ethics approval and consent to participate” sub-section as suggested by the reviewer. This is found on page 25 of the revised manuscript

We hope the revised manuscript will meet the standards for publication.

Thank you

Yours sincerely,

Dr. Kennedy Dodam Konlan

(Corresponding author)

---

## [Editor Report · Decision Letter 2]

2 Nov 2025

Supportive care needs and challenges experienced by women diagnosed with breast cancer in Kumasi, Ghana: A qualitative exploratory study

PONE-D-25-23079R2

Dear Dr. Konlan,

We’re pleased to inform you that your manuscript has been judged scientifically suitable for publication and will be formally accepted for publication once it meets all outstanding technical requirements.

Kind regards,

Frank Kyei-Arthur, Ph.D.

Academic Editor

PLOS ONE

Additional Editor Comments (optional):

I suggest that the authors should not abbreviate 'breast cancer' in the list of references. Therefore, they should replace "BC" with "breast cancer" in the list of references.
---

## [Editor Report · Acceptance letter]

PONE-D-25-23079R2

PLOS ONE

Dear Dr. Konlan,

I'm pleased to inform you that your manuscript has been deemed suitable for publication in PLOS ONE. Congratulations! Your manuscript is now being handed over to our production team.

Kind regards,

on behalf of

Dr. Frank Kyei-Arthur

Academic Editor

PLOS ONE